# The Extraction of Ocean Tidal Loading from ASAR Differential Interferograms

**DOI:** 10.3390/s20030632

**Published:** 2020-01-23

**Authors:** Wei Peng, Qijie Wang, Yunmeng Cao

**Affiliations:** 1School of Geosciences and Info-Physics, Central South University, Changsha 410083, China; pengweicsu@csu.edu.cn; 2Division of Physical Sciences and Engineering, King Abdullah University of Science and Technology, Jeddah 23955, Saudi Arabia; ymcmrs@gmail.com

**Keywords:** ocean tidal loading, ASAR differential interferograms, kinematic PPP tidal estimates, two-dimensional wavelet decomposition

## Abstract

The spatiotemporal crustal non-tectonic deformation caused by ocean tidal loading (OTL) can reach the centimeters scale in coastal land areas. The temporal variation of the site OTL displacements can be estimated by the global positioning system (GPS) technique, but its spatial variation needs to be further determined. In this paper, in order to analyze the spatial characteristics of the OTL displacements, we propose a multi-scale decomposition method based on signal spatial characteristics to derive the OTL displacements from differential interferometric synthetic aperture radar (D-InSAR) measurements. The method was tested using long-term advanced synthetic aperture radar (ASAR) data and GPS reference site data from the Los Angeles Basin in the United States, and we compared the results with the FES2014b tide model. The experimental results showed that the spatial function of the OTL displacements in an ASAR image can be represented as a higher-order polynomial function, and the spatial trends of the OTL displacements determined by the InSAR and the GPS techniques are basically consistent with the FES2014b tide model. The root-mean-square errors of the differences between the spatial OTL displacements of these two methods and the FES2014b tide model are less than 0.8 mm. The results indicate that the OTL displacement extracted from InSAR data can accurately reflect the spatial characteristics of the OTL effect, which will help to improve the spatial resolution and accuracy of the OTL displacement in coastal areas.

## 1. Introduction

Ocean tidal loading (OTL) is the elastic response of the solid Earth to the periodic mass redistribution of the ocean, and its effect on crustal deformation gradually weakens with distance from the coast. The spatiotemporal variations of the OTL displacements can be measured by the global positioning system (GPS) technique [1,2], very long baseline interferometry (VLBI) [3], a superconducting gravimeter (SG) [4], and the interferometric synthetic aperture radar (InSAR) [5]. The GPS, VLBI, and SG techniques have been proved effective in the estimation of the site OTL displacements [6,7,8]. However, the spatial variation characteristics of the OTL effect need a technique that can consider higher spatial resolution observations, such as InSAR measurements. 

The OTL effect needs to be corrected in a differential InSAR interferogram due to the fact its spatial variation can reach the cm/100km scale. Previous studies have demonstrated that OTL displacements in differential InSAR interferogram are non-linear and should be removed by the use of an ocean tide models [9,10]. In recent years, researchers have established a series of ocean tide models, such as NAO99b [11], FES2014b [12], EOT11a [13], and TPXO8 [14]. However, the inaccuracy of the ocean tide models in offshore areas can cause error in the estimation of actual OTL displacements [15]. Therefore, we propose to extract the OTL displacements from a differential InSAR interferogram to analyze their spatial characteristics. 

The OTL effect and other large-scale signals, such as the solid Earth tidal (SET) displacement and the atmospheric delay error related to topography and orbital error, will be mixed in the differential interferogram [16,17,18,19]. The wavelength of the SET displacement is one or two orders larger than the OTL effect, and its effect in an Advanced Synthetic Aperture Radar (ASAR) interferogram can be easily corrected using a SET correction model [20]. The trajectory of a SAR satellite platform is linear in the imaging process, and the orbital error between two revisits can be fitted by a bilinear function. The atmospheric correction products released by the generic atmospheric correction online service for InSAR (GACOS) can be used to correct the atmospheric delay errors, the correction of the topography related atmospheric delay error have been proved to be possible in the Southern California area, but the residual atmospheric turbulence signal still exists in the interferogram [21,22]. The other signals in the interferogram are mainly crustal tectonic deformation, the topographic residual phase, the residual atmospheric turbulence signal, the phase unwrapping error, and the spatiotemporal decorrelation error. The large-scale crustal tectonic deformation can be analyzed in advance by the precise point positioning (PPP) solutions [23]. The remaining signals and the OTL effect have different spatial characteristics. Therefore, the OTL displacements can be separated using image decomposition and reconstruction.

Wavelet analysis is a classical time-frequency analysis method, and the two-dimensional (2-D) wavelet is a dimensional extension of wavelet analysis, which has been widely applied in image decomposition [24,25,26]. The multi-scale frequency signals in the interferogram can be decomposed into sub-images with different spatial scales using 2-D wavelet decomposition, and the signal trend from near to the sea to inland can be reconstructed to obtain the higher spatial resolution OTL displacements.

## 2. Data Sets and Processing

### 2.1. InSAR and GPS Data

The determination of the spatial OTL displacements in the coastal area depends on the spatial density of the observations. There is an enormous amount of InSAR and GPS data available for the Southern California area of the United States, and the differential interferogram has high coherence in this area and can thus be used to determine the spatial variation of the OTL effect. We used 13 ENVISAT ASAR single look complex (SLC) images provided by the European Space Agency (ESA) as the experimental samples, with the time range of the ASAR data being from June 7, 2008, to December 19, 2009. The SAR imaging time was UTC 18:01:00, and the revisit period of the SAR satellite was 35 days. The range of the ASAR image covers the coastal areas of Southern California, as shown in Figure 1. In this range, Point1 is the farthest point and Point2 is the closest point to the coast. Moreover, 85 GPS reference stations were selected in the SAR image range and nearby areas. The RINEX files of the GPS stations were obtained from the Scripps Orbit and Permanent Array Center (SOPAC), with the time range being from January 1, 2008, to December 31, 2011. The distribution of the GPS stations is shown in Figure 1.

### 2.2. Data Processing

The 13 ASAR SLC images were processed in GAMMA software, following the procedures of image co-registration, interferogram generation, flat-earth phase removal, phase unwrapping, and geocoding [27,28]. The ASAR image of June 7, 2008 was regarded as the reference image to register other images, and twenty-seven differential interferograms were then generated from the 13 ASAR images according to the small baseline principle. The imaging dates of the master-slave images of the differential interferograms are shown in Figure 3. The shuttle radar topography mission-1 (SRTM-1) products was used to remove the topographic phase, and the minimum cost flow (MCF) algorithm was used for the phase unwrapping. 

The kinematic PPP coordinates of the GPS network were processed in the PPP mode of the Bernese processing engine (BPE) module of Bernese GPS software V5.2. The sample interval was 600s, and the input files of the satellite clock, precise ephemeris data, Earth rotation parameters, ionosphere maps, and differential code biases were provided by CODE (ftp://ftp.aiub.unibe.ch/CODE/). 

The SET effect was corrected in the data processing of the GPS and InSAR measurements, while OTL correction was not undertaken. 

## 3. Methodology

The observation at pixel i in an unwrapped interferogram can be expressed as:(1)$$d_{i,Unwrapped}=d_{i,Orb}+d_{i,Tidals}+d_{i,Atm}+d_{i,Tectonic\_Defo}+d_{i,Topo}+\varepsilon_{i}$$
where $d_{i,Orb}$ is the orbital-related error; $d_{i,Tidals}$ refers to the OTL and SET displacements; $d_{i,Atm}$ is the topography-related atmospheric delay error and the atmospheric turbulence error; $d_{i,Tectonic\_Defo}$ represents the tectonic crustal deformation; $d_{i,Topo}$ is the topographic residual phase; and $\varepsilon_{i}$ denotes the phase unwrapping error, the spatiotemporal decorrelation error, etc. 

The flowchart of the data processing and the extraction of OTL displacements from ASAR differential interferograms is shown in Figure 2.

### 3.1. Correction of the solid Earth tidal Effect, the Long-Wavelength Atmospheric Delay, and Orbital Error

The SET effect, and the large-scale spatial part of the atmospheric delay and orbital error can be removed by the commonly used correction models and functions. Firstly, the SET displacements were corrected by the SET correction model and the atmospheric delay error was corrected by the GACOS correction maps. Therefore, the pixel i in Equation (1) can be expressed as: (2)$$d_{i,Unwrapped-SET-GACOS}=d_{i,Orb}+d_{i,OTL}+d_{i,Atm\_resdidual}+d_{i,Tectonic\_Defo}+d_{i,Topo}+\varepsilon_{i}$$
where $d_{i,Unwrapped-SET-GACOS}$ represents the interferogram after the atmospheric delay error is corrected; $d_{i,OTL}$ is the OTL displacements; $d_{i,Atm\_resdidual}$ is mainly the residual atmospheric turbulence signal [21,22].

Secondly, the orbital error was estimated by bilinear function under the constraint of the kinematic PPP tidal estimates of the GPS network [29]. The site OTL displacements of the J GPS reference sites were used to represent the corresponding OTL displacements in the interferogram. Therefore, the expression of pixel j in the interferogram can be translated to:(3)$$d_{j,Unwrapped-SET-GACOS}=d_{j,Orb}+d_{j,OTL}^{GPS}+d_{i,Atm\_resdidual}+d_{j,Tectonic\_Defo}+d_{j,Topo}+\varepsilon_{j}$$
where $d_{j,OTL}^{GPS}$ represents the difference of the PPP tidal estimates at the imaging times of the master ($t_{master}$) and slave images ($t_{slave}$). 

The OTL displacements in the line of sight (LOS) direction of a SAR satellite can be modeled using a tidal harmonic function, where the semi-diurnal and diurnal tidal constituents *M*_2_, *N*_2_, *S*_2_, *K*_2_, *K*_1_, *O*_1_, *P*_1_, and *Q*_1_ are the main components of this OTL displacement time series [30]. Therefore, the PPP tidal estimates $d_{j,OTL}^{GPS}$ at the corresponding pixels in the differential interferogram can be expressed as: (4)$$d_{j,OTL}^{GPS}=\sum_{h=1}^{8}f_{j,h}A_{j,h}\left[\cos(\omega_{j,h}t_{master}+\chi_{j,h}+\mu_{j,h}-\Phi_{j,h})-\cos(\omega_{j,h}t_{slave}+\chi_{j,h}+\mu_{j,h}-\Phi_{j,h})\right]$$
where $A_{j,h}$ and $\Phi_{j,h}$ are the amplitude and phase lag of the hth tidal constituent, respectively; $\chi_{j,h}$ is the initial astronomical angle; $\omega_{j,h}$ is the angular frequency; and $f_{j,h}$ and $\mu_{j,h}$ are the node factor and the astronomical angle, respectively. 

Based on the spatial characteristics of the orbital error, the differential interferogram is fitted by a bilinear function under the constraint of the PPP tidal displacement $d_{j,OTL}^{GPS}$.
(5)$$d_{j,Unwrapped-SET-GACOS}-d_{j,OTL}^{GPS}=a_{0}+ax_{j}+by_{j}+cx_{j}y_{j}$$
where $x_{j}$, $y_{j}$ represent the location of the jth GPS site. We solve coefficients $a_{0}$, a, b, and c to estimate the orbital errors $d_{i,Orb}$ in the interferograms.

Finally, $d_{i,Orb}$ was substituted into Equation (2) to eliminate orbital error. After the SET effect, orbital error and long-wavelength atmospheric delay are eliminated, Equation (2) can be converted to:(6)$$d_{i,Unwrapped-SET-GACOS-orbital}=d_{i,OTL}+d_{i,Atm\_turb\_resdidual}+d_{i,Tectonic\_Defo}+d_{i,topo}+\varepsilon_{i}$$

According to the prior information of the tide model, the spatial wavelength scale of the OTL signal is more than 10^2^ km. The site velocities of the GPS network indicated that the interferograms used in this study did not contain large-scale tectonic deformation (Figure A1). Therefore, In Equation (6), the spatial scale of the residual atmospheric turbulence error $d_{i,Atm\_turb\_resdidual}$, the crustal structural deformation $d_{i,Tectonic\_Defo}$, residual topography phase $d_{i,topo}$, the phase unwrapping error and spatiotemporal decorrelation error $\varepsilon_{i}$ were smaller than that of the OTL signal. The two-dimensional wavelet decomposition is based on the different spatial scales of the sub-signals. Therefore, the large spatial scale OTL signal in the differential interferogram $d_{i,Unwrapped-SET-GACOS-orbital}$ could be separated from the other signals using this multi-scale image decomposition method.

### 3.2. Multi-Scale Decomposition Using the 2-D Wavelet Transform Method

Two-dimensional wavelet decomposition is a dimensional expansion based on the principle of wavelet transform, and it decomposes the row and column of 2-D signals sequentially using one-dimensional wavelet decomposition. The matrix of an M-row and N-column interferogram $d_{i,Unwrapped-SET-GACOS-orbital}$ can be decomposed by the 2-D wavelet decomposition into the approximation coefficients $A_{k}$ and the horizontal $D_{k}^{H}$, vertical $D_{k}^{V}$, and diagonal detail coefficients $D_{k}^{D}$.
(7)$$A_{k,m,n}=\frac{1}{\sqrt{MN}}\sum_{x^{\prime }=0}^{M-1}\sum_{y^{\prime }=0}^{N-1}f(x^{\prime },y^{\prime })\phi_{k,m}(x^{\prime })\phi_{k,n}(y^{\prime })\;D_{k,m,n}^{H}=\frac{1}{\sqrt{MN}}\sum_{x^{\prime }=0}^{M-1}\sum_{y^{\prime }=0}^{N-1}f(x^{\prime },y^{\prime })\phi_{k,m}(x^{\prime })\psi_{k,n}(y^{\prime })\;D_{k,m,n}^{V}=\frac{1}{\sqrt{MN}}\sum_{x^{\prime }=0}^{M-1}\sum_{y^{\prime }=0}^{N-1}f(x^{\prime },y^{\prime })\psi_{k,m}(x^{\prime })\phi_{k,n}(y^{\prime })\;D_{k,m,n}^{D}=\frac{1}{\sqrt{MN}}\sum_{x^{\prime }=0}^{M-1}\sum_{y^{\prime }=0}^{N-1}f(x^{\prime },y^{\prime })\psi_{k,m}(x^{\prime })\psi_{k,n}(y^{\prime })$$
where $\left(x^{\prime },y^{\prime }\right)$ is the row and column number of the image pixel points, k is the wavelet level, ϕ is a scale function, and ψ is the wavelet basis. 

We reconstruct the wavelet decomposition components to extract the implied source signal. The coefficients are organized as a vector C:(8)$$C=[A_{k},D_{k},D_{k-1},\cdots ,D_{1}]$$

The detail coefficients $D_{k},D_{k-1},\cdots ,D_{1}$ are set to zero matrices for extracting the kth wavelet level reconstructed signal. A higher wavelet level k can obtain the larger spatial scale of the reconstructed signal. Based on the prior information of the OTL displacement of the tide model, the wavelet level k is determined to reconstruct the wavelet decomposition components into OTL displacements.

## 4. The Ocean Tidal Loading Effect in a D-InSAR Interferogram

### 4.1. The OTL Displacements Calculated Based on FES2014b Tide Model

The degree of spatial deformation caused by the OTL effect in the SAR images was preliminary analyzed using the FES2014b tide model. The relative spatial variations of the OTL displacements between the near coastline point Point2 and inland point Point1 (Figure 1) are shown in Table 1. 

From the results shown in Table 1, it can be seen that the maximum and minimum values are 4.4 mm and −6.5 mm. Therefore, the OTL displacements in the interferograms generated by these two ASAR images are over 10 mm, which needs to be corrected. Furthermore, the OTL displacements in the 27 differential interferograms were calculated based on the FES2014b tide model are shown in Figure 3.

The spatiotemporal OTL variations of most of the differential interferograms are less than 6 mm, but the maximum difference between the relative spatial variation of the OTL displacement at the imaging dates of 29 November 2008 and 18 April 2019 can reach 10.9 mm, and that of the imaging dates of 18 April 2019 and 19 Devember 2009 is 10.8 mm. We selected the two interferograms with the stronger spatiotemporal OTL variations for further analysis of the kinematic PPP tidal estimates and differential interferogram decomposition.

### 4.2. The OTL Displacements Estimated by Kinematic PPP

The OTL displacements of the eight main tidal constituents (semi-diurnal: *M*_2_, *S*_2_, *N*_2_, *K*_2_; diurnal: *K*_1_, *O*_1_, *P*_1_, *Q*_1_) at the GPS sites can be calculated according to the load Love numbers driven Green’s function [31]. The PPP tidal estimates for the differential interferograms are then generated by Equation (4). The spatial variation of the PPP tidal displacement is shown in Figure 4, and the comparative analysis with the FES2014b model is shown in Figure 5.

In Figure 4, the PPP tidal estimates show a variation trend moving from coast to inland. Moreover, the straight slopes of the PPP tidal displacements agree with the FES2014b model tidal estimates in Figure 5, and the misfits can be regarded as random errors and outliers. The RMSE values are 4.07 and 4.77 mm. 

### 4.3. The OTL Displacements Extracted From the Differential Interferograms

The extraction of the OTL displacements from ASAR differential interferograms needs to correct the large-scale spatial signals. The SET corrections for the two selected differential interferograms are 2.1 and 1.1 mm, which are shown in Figure 6.

The atmospheric delay error in the differential interferogram was corrected by the use of the GACOS atmospheric products. The pixels of the interferograms corresponding to the PPP tidal estimates were then selected for estimating the orbital error, and we used the PPP tidal estimates as the a *priori* information, so that the OTL displacements were not removed as “orbital error” from the differential interferogram after the atmospheric delay error correction.

In the final results shown in Figure 7, the small-scale spatial signal in the interferograms is mainly the residual atmospheric turbulence signal, and the trend from near coast to inland still exist in the interferograms. Therefore, the two-dimensional wavelet method was used to decompose these interferograms.

There are many wavelet basis families, such as coif wavelets and sym wavelets, which can be used to extract the OTL signal in an interferogram. We chose the coif5 wavelet basis to decompose the interferograms into sub-images, according to the result comparison and previous research [25,26]. The sub-images of the different wavelet scales decomposed from the differential interferograms after atmospheric delay and orbital error correction are shown in Figure 8.

The spatial function of the OTL displacements extracted from the interferograms is close to a higher-order polynomial.
(9)$$d_{i,OTL}=p_{0}+{\left(px+qy\right)}^{r}$$
where x and y are the longitude and latitude of pixel i in the interferogram. r is the order of the polynomial. $p_{0}$, p, and q are the polynomial coefficients, which vary with the spatial variation of the OTL displacement.

We also modeled the PPP tidal estimates using the higher-order polynomial. The selection of the orders was based on the ocean tide model and OTL displacements extracted from the interferograms, which turned out to be three and four orders. The OTL displacements derived using the 2-D wavelet decomposition and the modeled PPP tidal estimates are compared with the FES2014b tide model, which are shown in Figure 9.

In Figure 9, the trend variations of the extracted OTL displacement and the modeled PPP tidal estimates are basically consistent with FES2014b model tidal estimates. For differential interferogram for 29 November 2008 to 18 April 2009, the root-mean-square error (RMSE) of the misfit between the extracted OTL displacements and the FES2014b model tidal estimates is 0.28 mm, and that of the modeled PPP tidal estimates and the FES2014b model tidal estimates is 0.69 mm. The RMSEs for the differential interferogram for 18 April 2009 to 29 November 2009 are 0.76 and 0.52 mm, respectively. Compared with the FES2014 tidal model, the extracted OTL displacement and the modeled PPP tidal estimates show that the spatial OTL approach to higher order polynomials in the near sea area. However, the extracted OTL displacements for 18 April 2009 to 29 November 2009 appear a larger difference in the inland area, which is considered as the multiscale decomposition affected by the residual atmospheric signal, because the tectonic deformation (Figure A1), SET effect and orbital error in our study would not introduce this regional error.

## 5. Conclusions

In this paper, we have described how ocean tide models, regional GPS network data, and differential interferogram decomposition can be applied to the analysis of the OTL effect in interferograms. In the study of the selected interferograms from Southern California, the OTL effect and the long-wavelength errors were separated, and the spatial characteristic of the OTL effect in the interferogram was considered as a higher-order polynomial, with the order varying in the different interferograms. For the two selected interferograms, the RMSEs of the extracted OTL displacements and the modeled PPP tidal estimates, with regard to the FES2014b model tidal estimates, were less than 0.7 and 0.8 mm, indicating that it is possible to determine the spatiotemporal OTL effect in differential interferograms. By determining the OTL displacement from ASAR data, the spatial resolution of the OTL displacement can be improved, and the spatial characteristics of the OTL effect in the differential interferograms can be further determined.

## Figures and Tables

**Figure 1 sensors-20-00632-f001:**
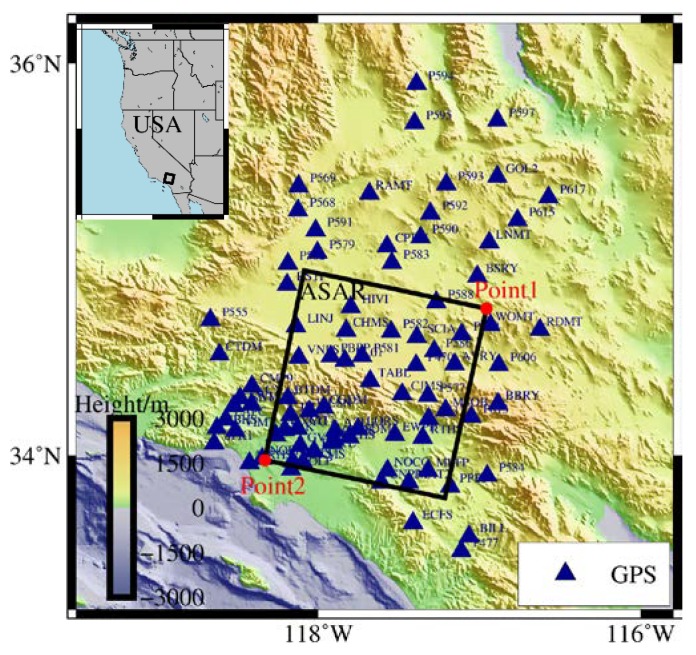
The image range of the Advanced Synthetic Aperture Radar (ASAR) data (black box) and the distribution of the global positioning system (GPS) sites (dark blue triangles). Point1 is the farthest point and Point2 (red dots) is the closest point to the coast in the synthetic aperture radar (SAR) image.

**Figure 2 sensors-20-00632-f002:**
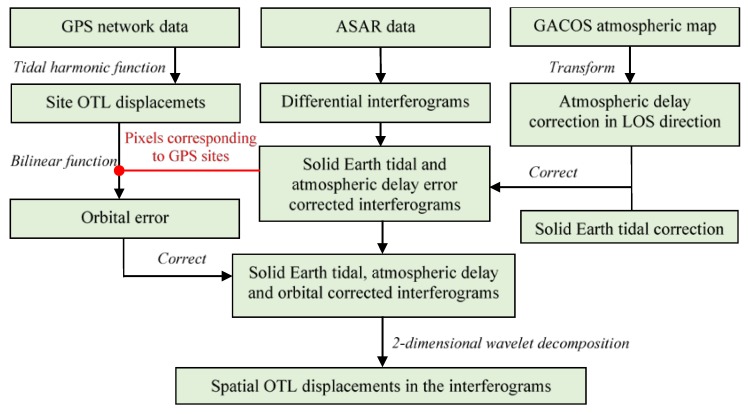
Flowchart of the extraction of Ocean tidal loading (OTL) displacements from ASAR differential interferograms.

**Figure 3 sensors-20-00632-f003:**
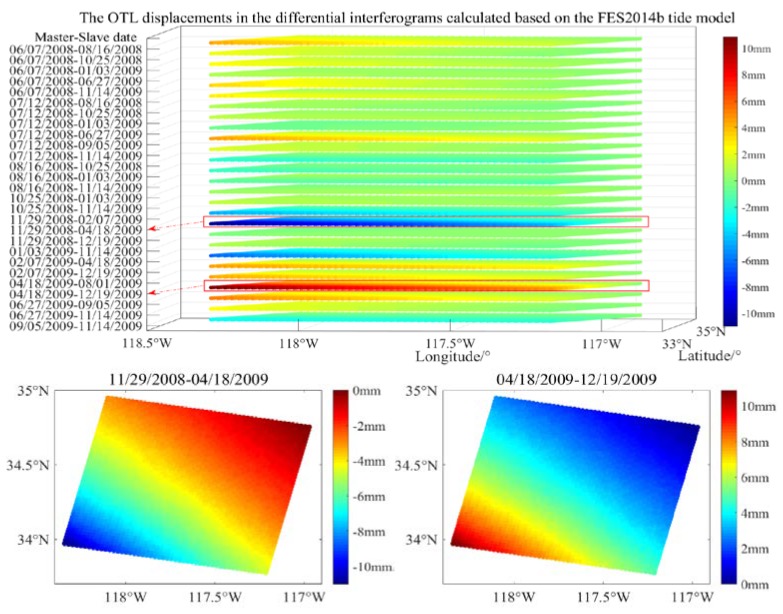
The spatiotemporal OTL effect in the 27 differential interferograms based on the FES2014b model, and the two selected interferograms for details of the spatial OTL displacements over 10 mm.

**Figure 4 sensors-20-00632-f004:**
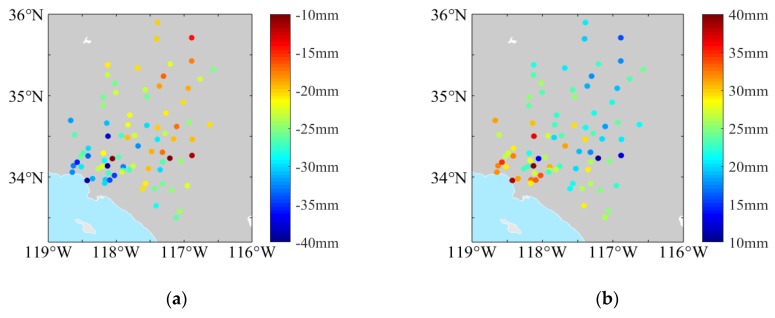
The relative variation of the site OTL displacements for analyzing the OTL effect in (**a**) interferogram acquisition for 29 November 2008 to 18 April 2009, and (**b**) interferogram for 18 April 2009 to 19 Devember 2009.

**Figure 5 sensors-20-00632-f005:**
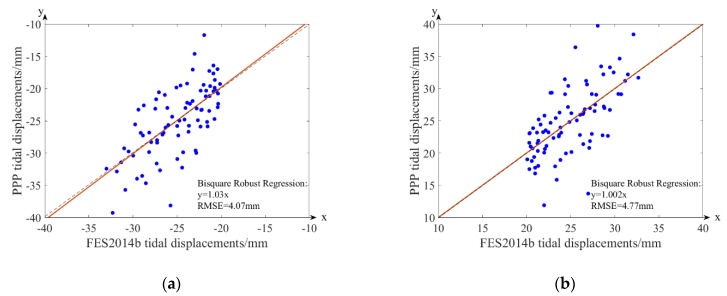
Comparative analysis of the site OTL displacements and the FES2014b tide model in (**a**) interferogram for 29 November 2008 to 18 April 2009, and (**b**) interferogram for 18 April 2009 to 19 Devember 2009.

**Figure 6 sensors-20-00632-f006:**
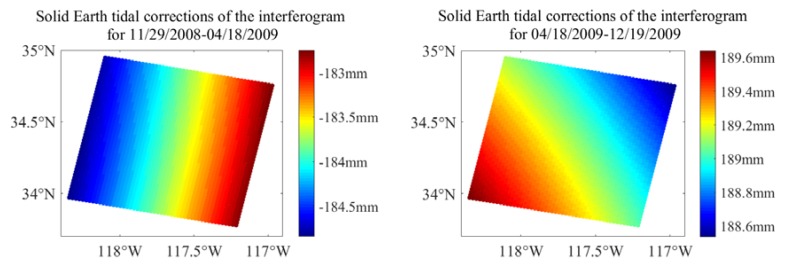
The solid Earth tidal (SET) correction of the interferogram for 29 November 2008 to 18 April 2009 and 18 April 2009 to 19 Devember 2008.

**Figure 7 sensors-20-00632-f007:**
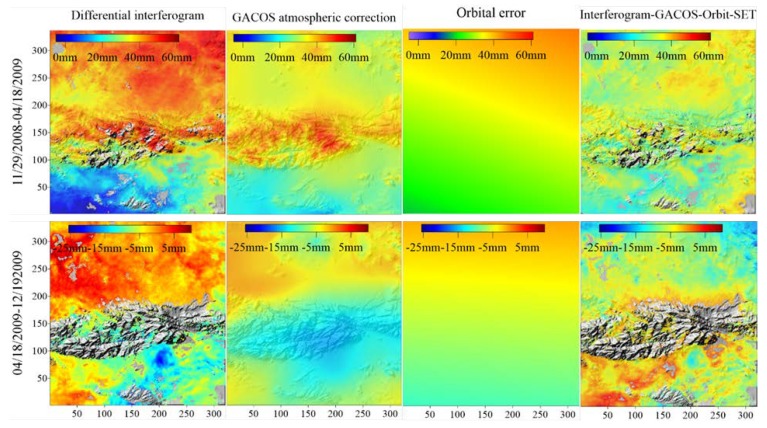
Comparison of the differential interferogram, the generic atmospheric correction online service for InSAR (GACOS) atmospheric map, the orbital error constrained by precise point positioning (PPP) tidal estimates, and the differential interferogram after SET, atmospheric delay and orbital error correction.

**Figure 8 sensors-20-00632-f008:**
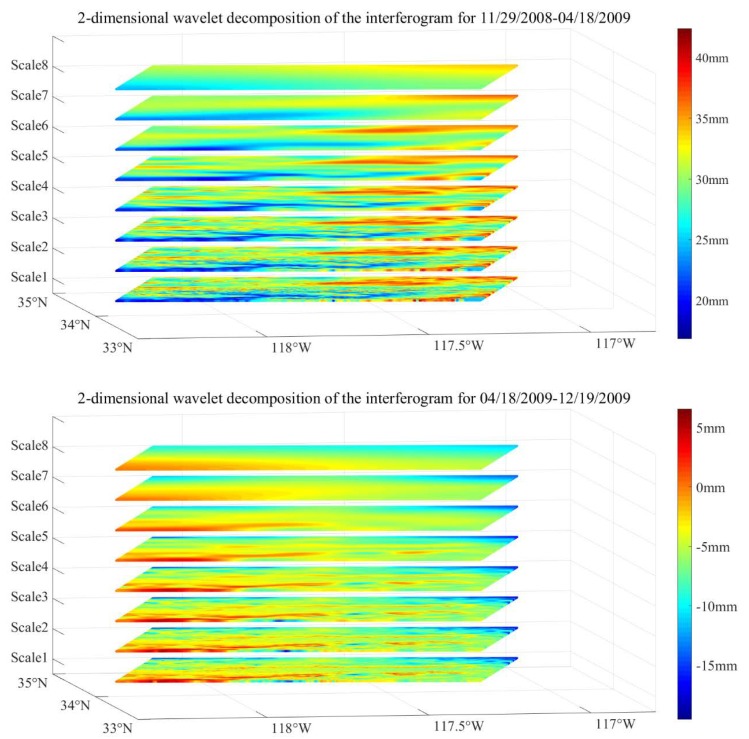
The sub-images of different wavelet scales decomposed from the differential interferograms after atmospheric delay and orbital error correction.

**Figure 9 sensors-20-00632-f009:**
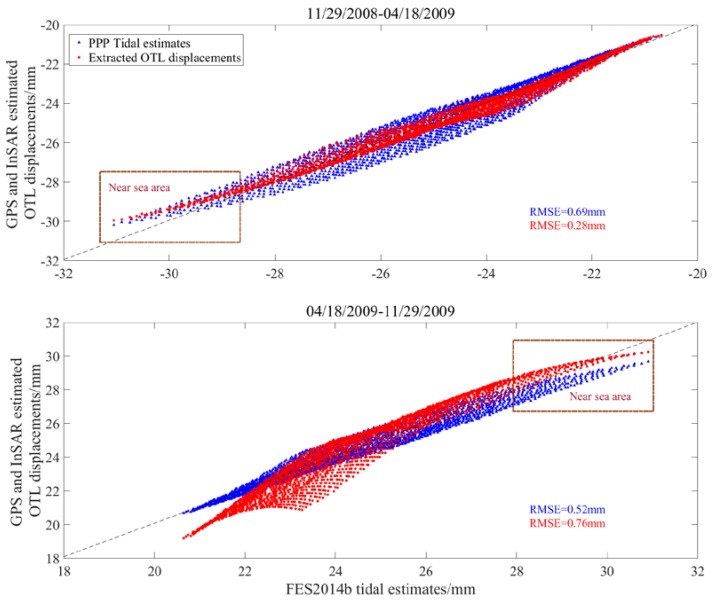
Comparison of the extracted OTL displacements and the modeled PPP tidal estimates with the FES2014b model tidal estimates in the interferogram for 11/29/2008 to 04/18/2009, and the interferogram for 04/18/2009 to 12/19/2009.

**Table 1 sensors-20-00632-t001:** Relative spatial variation of the OTL displacement in the LOS direction at imaging time 18:01:00 UTC for the near/far coastline points (Point2 /Point1 ) in the 13 ASAR images based on the FES2014b tide model.

Imaging Date	Relative Spatial Variation of the OTL Displacement/mm	Imaging Date	Relative Spatial Variation of the OTL Displacement/mm
7 June 2008	2.4	18 April 2009	4.4
12 July 2008	0.9	27 June 2009	1.6
16 August 2008	−1.8	1 August 2009	0.3
25 October 2008	0.2	5 September 2009	−3.7
29 November 2008	−6.5	14 November 2009	−0.9
3 January 2009	−0.2	19 December 2009	−6.4
7 February 2009	−1.8

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
