# Peer review of "The Extraction of Ocean Tidal Loading from ASAR Differential Interferograms"

_sensors, 2020, doi:10.3390/s20030632_

Round 1
Reviewer 1 Report
This paper extracts OTL displacements from GPS, InSAR, and tidal models. It is an interesting paper. I recommend this paper for publication after minor revision. I would like to suggest the authors use a flowchart to present their procedures unambiguously. Major and minor comments are listed below.
Section 2.2 mentioned that the solid Earth tidal effect was corrected in the data processing. Are the corrections made for both the ASAR interferogram and the kinematic PPP coordinates. If the corrections are made for ASAR interferogram, could you please explain a bit more how this is implemented and how much solid Earth tidal effect is corrected ? If the corrections are not made for InSAR, could you please justify why Equation (1) ignored the solid Earth tidal effect in an unwrapped interferogram ?
Section 3 concluded that the spatial scale of the tectonic crustal deformation is smaller than that of the OTL signal. Could you please give a reference to support this ? Or quantitatively, what is the upper limit of the spatial scale for a tectonic crustal deformation in order not to obscure OTL signals ?
Figure 4, for the interferograms after the correction of the long-wavelength atmospheric delay error and 200 orbital error, have you used Equation (3) to subtract PPP tidal displacement before bilinear estimation for orbital error as seen from Equation (4)? If PPP tidal displacement are already subtracted from the interferograms, why they are still used to extract OTL signals ?
Minor comments:
Line 107, pixel index is missing.
Line 118 and 119, pixel index i is referred in Line 118, but not used in Equation (2).
Line 127, In Equation (3), does j refer to GPS site or the eight tidal constituents M2, N2, …, Q1. If j is used to represent main OTL components in Equation (3), I would like the authors to be consistent with their indexing as j is referred to GPS site in Equation (2) and (4). If j is solely referred to GPS site in the manuscript, please ignore my question.
Line 140, ε is defined as spatiotemporal error in Line 140, but it is defined as phase unwrapping error in Line 110 and 111.
Line 147, what does M and N stand for ? M pixels, N interferograms, for multiple interferograms or M line, N columns, for a single interferogram ?
Reviewer 2 Report
Review Sensors-678005
Thank you very much for the opportunity to review this article. I have enjoyed it very much. The paper proposes a new way to extract OTL from ASAR differential Interferograms. While it is promising, I have some major concerns with the manuscript as it is now, especially on regards to the format and clarity of the assumptions. You can find some of my suggestions below:
Abstract – From the abstract itself, it is not clear what the main objective of your research is. It is not clear what is known, what is not known, and what is the novelty of your research. This should be clear. Without an objective, it is hard to follow your research. Also, there is information stated here that becomes clear only after reading the article, which is not proper for an abstract (e.g. FES2014 reference on line 26)
Introduction – Here, besides some unclear sentences, the main point is the lack of direction of the structure itself. I finished the introduction without a clear definition of the problem, without a clear idea of what the paper is about precisely, and without really knowing why this study was done. It is imperative to use the introduction to show what is known, what is not known, and what you bring with your article. Besides, some paragraphs should be placed in methodology (e.g. line 69), double citations should be removed (e.g. line 38), and tons of information should be properly organized. I strongly suggest a rewriting of this section. Make sure that these questions are answered: What is known? What is not known? What am I going to do? Why this is important?
Data Sets and processing:
Line 84 to 85: Why is it valid to use this area as a case study? Why can this be representative of all the other regions?
Line 86 to 88: Why did you choose those 13 ENVISAT ASAR images?
Figure 1: The figure needs a proper legend, scale and colour scale.
Line 98: Be specific on what did you do with the images (research reproducibility). It is far more essential to derive what the software has done instead of citing it, even when it is widely known inside the community.
Methodology:
Line 113: Why did you use this assumption if you already stated that it might be poor (line 65) without further testing? In other words, why is this assumption valid?
Line 116: How did you exactly correct the images, and how this correction relates explicitly to the assumptions?
Section 4:
Figure 2: Figure is not clear in the current state. Please redo.
Line 177: How did you statistically test for significance?
Figure 3: Do not separate the same figure in different pages. Besides, fonts are too small, and the whole figure is not clear. Furthermore, any trend line should be plotted with its function and the associated error/Squared R.
Figure 4: The Figure is also not fully readable. Colour scales can be improved. Also, I suggest changing the colour scale for different variables. Also, please add a reference to Y-axis and X-axis.
Figure 6: The figure is not clear. Also, the text says that it is a comparison, but it is not clear what is the comparison you want to do exactly. Just plotting does not necessarily mean you are comparing something. Please, help the reader and explain what you want to show. Lead the story and show what you want to show.
In conclusion, I think the paper has potential, but it needs some major revisions in the writing aspect for further publication. The main problem is the lack of direction of the paper regarding structure, problem definition, clarity and precision. Nonetheless, I wish to acknowledge the authors for the hard work, and I hope my comments help on improving this manuscript further — also, best wishes for this new year.
Round 2
Reviewer 1 Report
The authors have addressed all my concerns and I have no more questions.
Reviewer 2 Report
Dear Authors,
First of all, thank you very much for your kindness in the review processes. The rebuttal letter was clear and I am satisfied with the answers. The new version is much improved in terms of clarity and problem definition, which was the main problem with the previous version, in my opinion. There are still some parts of the text that are not clear enough due to, for example, the use of passive voice and some ambiguity, but that do not affect the overall message of the paper. I wish all the best for the authors and I am looking forward to reading new research from the group.
Best Regards